# Spurious North Tropical Atlantic precursors to El Niño

Wenjun Zhang [1]✉, Feng Jiang[1], Malte F. Stuecker [2], Fei-Fei Jin[3]✉ & Axel Timmermann[4,5]

The El Niño-Southern Oscillation (ENSO), the primary driver of year-to-year global climate variability, is known to influence the North Tropical Atlantic (NTA) sea surface temperature (SST), especially during boreal spring season. Focusing on statistical lead-lag relationships, previous studies have proposed that interannual NTA SST variability can also feed back on ENSO in a predictable manner. However, these studies did not properly account for ENSO's autocorrelation and the fact that the SST in the Atlantic and Pacific, as well as their inter-action are seasonally modulated. This can lead to misinterpretations of causality and the spurious identification of Atlantic precursors for ENSO. Revisiting this issue under con-sideration of seasonality, time-varying ENSO frequency, and greenhouse warming, we demonstrate that the cross-correlation characteristics between NTA SST and ENSO, are consistent with a one-way Pacific to Atlantic forcing, even though the interpretation of lead-lag relationships may suggest otherwise.

[1] Key Laboratory of Meteorological Disaster of Ministry of Education (KLME), Nanjing University of Information Science and Technology, Nanjing, China. [2] Department of Oceanography & International Pacific Research Center (IPRC), School of Ocean and Earth Science and Technology (SOEST), University of Hawai'i at Mānoa, Honolulu, HI, USA. [3] Department of Atmospheric Sciences, School of Ocean and Earth Science and Technology (SOEST), University of Hawai'i at Mānoa, Honolulu, HI, USA. [4] Institute for Basic Science, Center for Climate Physics, Busan, South Korea. [5] Pusan National University, Busan, South Korea. ✉email: zhangwj@nuist.edu.cn; jff@hawaii.edu

The El Niño–Southern Oscillation (ENSO) phenomenon is characterized by interannual fluctuations between warm (El Niño) and cold (La Niña) sea surface temperature (SST) conditions in the equatorial Pacific. Its dynamics and associated coupled changes in the atmosphere and ocean have been studied extensively[1,2]. Conceptual frameworks for ENSO have been proposed to explain the statistical and physical characteristics in terms of a Pacific eigenoscillation that originates from positive air–sea interactions and delayed oceanic negative feedbacks[3–6]. ENSO is further energized by stochastic atmospheric forcing[7] and modulated by the seasonal cycle[8,9]. Counterintuitively, despite significant advances in both ENSO theory and ENSO representation in climate models, the predictability of central-to-eastern tropical Pacific SST anomalies has decreased in the past two decades to only one season[10–12]. Research over the past years has further revealed that SST anomalies in other ocean basins may also play an important role shaping the evolution of El Niño events and its predictability[13–24]. In particular, the North Tropical Atlantic (NTA) SST has been highlighted as a potential precursor candidate[18,22–24].

The NTA ocean, home to a variety of societally relevant climate phenomena has received widespread attention[25–29]. Typically, NTA SST warming lags the El Niño mature winter phase, peaking in the following spring (Fig. 1a) and persisting into early summer[30]. It is caused by El Niño-induced atmospheric forcing that both modulates the Walker Circulation and excites the Pacific-North America teleconnection pattern[31–35]. In turn, this NTA warming is argued to stimulate a westward-propagating off-equatorial Rossby wave train, conducive to an ensuing La Niña formation[18,24]. However, this reverse connection, which is characterized by a negative ENSO-NTA cross-correlation with the NTA SST leading by about 8 months (Fig. 1b), is highly variable and especially absent before the 1990s[22] (Fig. 1c). Despite some presumptions involved[22,36], the mechanisms responsible for the puzzling connection are less appreciated and a comprehensive understanding of the two-way interaction between NTA variability and ENSO is required.

In this study, we use both observations and climate model simulations to interpret this time-varying relationship. We demonstrate that changes in the NTA-ENSO relationship can be explained in terms of changes in ENSO frequency. The proposed mechanism is consistent with ENSO forcing NTA, rather than the opposite.

## Results

### ENSO-NTA SST relationship in observations.
ENSO generally commences its development in boreal summer and peaks in winter, stimulating atmospheric forcing over the NTA through two distinct pathways involving tropical and extra-tropical teleconnections[30,34,35]. Analyzing observed SST anomalies (see Methods), we see that the El Niño remote forcing is felt in the NTA SST 3–5 months later around the spring season (Fig. 1a), possibly due to the local SST adjustment timescale[37] and the seasonality of the atmospheric teleconnection to the Atlantic[30,35]. This robust ENSO-NTA connection can be detected during the entire study period notwithstanding a slight reduction of the correlation coefficient in the recent two decades (red shading in Fig. 1c; see also ref. [38]). In turn, the spring NTA warming appears to contribute to the following La Niña development in the Pacific Ocean (and similarly a spring NTA cooling contributing to a following El Niño) (Fig. 1b) with a relatively weak correlation at about 8-month lag. However, we must also emphasize here that an NTA warming in spring following an El Niño will automatically be correlated with La Niña conditions 8 months later, because El Niño conditions are usually followed by La Niña in the

following year, without involving a physical NTA-to-ENSO relationship. Therefore, one needs to be careful in interpreting seasonally modulated teleconnections of ENSO (see for instance discussion in ref. [21]). The 8-month leading relationship of NTA over ENSO is observed after the early 1990s while it is absent in the preceding period (blue shading in Fig. 1c; see also ref. [22]). Prior to the 1990s we find a much longer characteristic lead of ~20-month (blue shading in Fig. 1c). Interestingly, this decadal change of the NTA-lead-time corresponds well to a shift in ENSO frequency from quasi-quadrennial to quasi-biennial (Supplementary Fig. 1; see also ref. [39]). This regime change is also accompanied by more frequent occurrences of Central Pacific (CP) ENSO events (characterized by quasi-biennial timescale) and a reduction of the canonical Eastern Pacific (EP) ENSO events (characterized by quasi-quadrennial timescale)[2,12].

Here we hypothesize that the changing ENSO-NTA SST phase-lag relationships can be explained in the context of different ENSO regimes manifested by quasi-biennial and quasi-quadrennial periodicities. An El Niño is typically followed by a La Niña event during the subsequent winter in a quasi-biennial ENSO cycle, whereby the NTA warming in the decaying El Niño spring accompanies a La Niña formation about 8-month later. For a quasi-quadrennial ENSO cycle, it takes around two years for the phase transition on average and correspondingly an El Niño induced NTA warming statistically leads the next La Niña mature phase by about 20 months.

To further illustrate the abovementioned physical linkage between lead time and ENSO frequency, we conduct 2–3- and 3–5-yr bandpass filtering of the observed ENSO and NTA indices to differentiate two-way ENSO-NTA SST connections associated with quasi-biennial and quasi-quadrennial periodicities, respectively. ENSO impacts on boreal spring NTA SST anomalies are clearly displayed in both ENSO frequency bands (Fig. 1d), consistent with the robust relationship derived from the raw data (red shading in Fig. 1c), substantiating ENSO's physical regulation of the following spring NTA SST. To understand the distinct statistical relationships of the NTA SST with quasi-biennial and quasi-quadrennial ENSO (negative lags in Fig. 1c), we need to consider first that for these timescales El Niño and La Niña are anticorrelated at a lag of ~12 months and ~24 months, respectively. With El Niño causing robust spring NTA warming, the spring NTA warming will then be automatically anticorrelated with Niño3.4 SST anomalies at lag 8 (=12−4) and lag 20 (=24−4) months, for the quasi-biennial and quasi-quadrennial modes, respectively (Fig. 1d). The decadal shifts in the NTA-ENSO relationship are thus consistent with a robust one-way ENSO to NTA forcing relationship combined with a shift of ENSO's dominant frequency (Supplementary Fig. 1b).

Next, to understand the role of ENSO forcing in fostering NTA variability when considering its time-varying periodicity change, we use an extension of the original stochastic climate model[40] for NTA SST anomalies that includes both remote observed ENSO forcing and a damping rate modulated by the annual cycle (see Methods and ref. [21] for the original application of the model). The observed monthly time-varying NTA SST anomaly can be well captured by the ENSO-forced model ($R = 0.55$, statistically significant at the 95% confidence level; Supplementary Fig. 2). Importantly, the residual variability has no preferred interannual spectral peak (Supplementary Fig. 3). The reconstructed NTA SST exhibits a very similar lead-lag relationship with ENSO compared to that of the observations (Fig. 1d), further collaborating our hypothesis of a one-way relationship between the tropical Pacific and North Atlantic climate variability. Note that observed ENSO cycles are not perfect oscillations with a distinct periodicity, as in the case of the quasi-quadrennial cycle in which a strong El Niño event is prone to be followed by

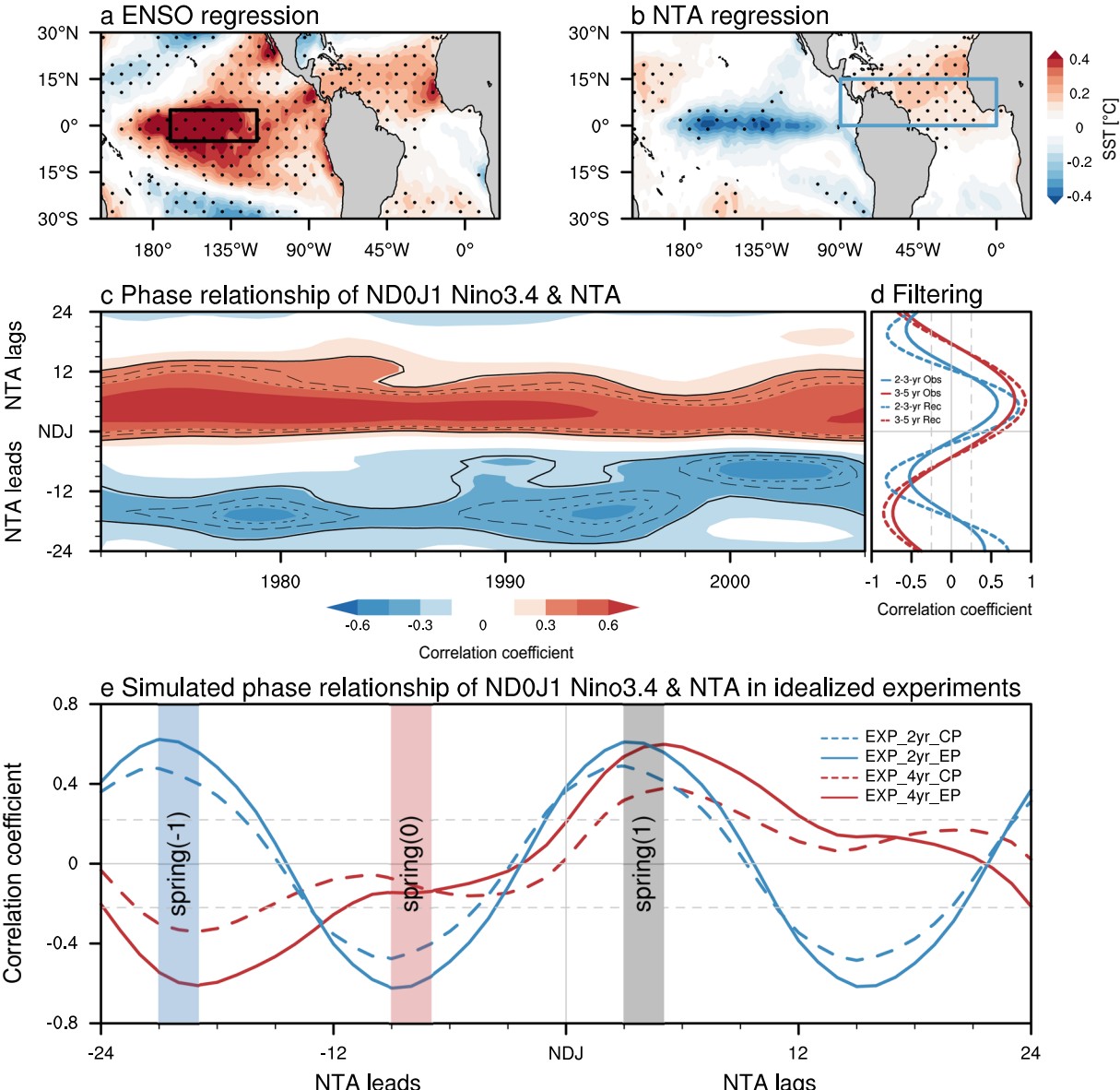

**Fig. 1 Relationships between tropical Pacific and North Atlantic climate variability.** Regression of **a** boreal spring (March–May) sea surface temperature (SST) anomalies (shading; °C) upon the normalized preceding winter (November–January) Niño3.4 (black box; 5°S–5°N, 120°–170°W) index and **b** boreal winter SST anomalies (shading; °C) upon the normalized preceding spring North Tropical Atlantic (NTA) (blue box; 0°–15°N, 90°–0°W) SST anomaly. Dots in (a-b) indicate regression coefficients that are statistically significant at the 95% confidence level. **c** 15-yr running lead-lagged correlation of the boreal winter Niño3.4 index with the NTA SST anomaly. Solid, dashed, and dotted lines mark the region with values exceeding the 80, 90, and 95% confidence levels, respectively. **d** Lead-lagged correlation of the boreal winter Niño3.4 index with the observed (solid) and reconstructed (dashed) NTA SST anomaly for bandpass filtering of 2–3-yr (blue) and 3–5-yr (red) periods by using a Fast Fourier Transform filter. For the y-axis of **c–d**, negative and positive values indicate NTA-lead and NTA-lag at monthly scale, respectively. **e** Lead-lagged correlation of the boreal winter Niño3.4 index with NTA SST anomaly in the idealized pacemaker experiments with different Pacific SST forcing (see Methods). Gray dashed lines in **d–e** indicate the 95% confidence levels.

consecutive La Niña events. However, the complicated ENSO cycle features seemingly play a minor role in determining the relationship of NTA SST with following ENSO from a statistical standpoint (Supplementary Discussion and Supplementary Fig. 4).

**ENSO-NTA SST relationship in idealized pacemaker experiments.** Observed ENSO variability has a broad spectrum in the range of 2–7 years, characterized by quasi-biennial and quasi-quadrennial spectral peaks, which cannot be completely isolated using current linear methods[2]. To demonstrate trans-basin

relationships that would result from different purely periodic ENSO oscillations, a set of idealized pacemaker experiments is conducted by imposing ENSO SST anomaly forcing with idealized 2- and 4-yr cycles in the tropical Pacific (see Methods). In this modeling set-up, only ENSO can force NTA, but not vice versa. Given that there is a shift in the ENSO's zonal location around the 1990s, we also consider different SST forcing patterns associated with the EP and CP El Niño types in the pacemaker experiments (Supplementary Fig. 5; see Methods), to investigate possible influences of the zonal SST anomaly structure in addition to ENSO timescale changes. The observed robust ENSO effect on

the subsequent spring NTA SST can be well reproduced in all ENSO-forced experiments (Fig. 1e). In the experiments with 2-yr ENSO forcing, the NTA SST variability is significantly correlated with subsequent ENSO conditions of opposite sign, having the maximum correlation at an 8-month lead-time of NTA over ENSO regardless of the ENSO SST anomaly patterns. This statistical ENSO-NTA relationship corresponds to what we see in the observations after the 1990s (Fig. 1c). Our results clearly show that the 8-month lead of NTA over ENSO can be obtained, even though the set-up of our model experiments does not allow for NTA to influence ENSO. In the 4-yr ENSO forced experiments, the spring NTA SST anomaly as a response to the preceding ENSO is followed by the subsequent ENSO formation at about 20-month lead time for both EP and CP associated SST forcing. Slightly weaker ENSO-NTA lead-lag relationship is detected for 4-yr experiments versus 2-yr experiments, possibly due to our modeling set-up in which the idealized 4-yr ENSO cycle experiences longer time evolution of ENSO transition. These pacemaker experiments indicate that the statistical ENSO and NTA relationship is largely controlled by ENSO periodicity rather than its spatial pattern and that the ENSO auto-correlation itself causes this peculiar phase-relationship.

**ENSO-NTA SST relationship in the CMIP6 simulations.** Considering the limited sample size of the short observational record though supported by our idealized pacemaker experiments, we further examine the trans-basin relationship between ENSO and NTA SST in 46 coupled models in pre-industrial control (pi-control) simulations participating in Phase 6 of the Coupled Model Inter-comparison Project (CMIP6) (Supplementary Table 2). Almost all coupled models are capable of capturing the robust ENSO forcing on the NTA SST (Supplementary Fig. 6). The multi-model ensemble (MME) mean can reasonably reproduce the observed relationship between boreal winter ENSO and following spring NTA SST variability with a certain inter-model spread. However, the models exhibit a large diversity in the statistical relationship between boreal spring NTA SST variability and subsequent winter ENSO at ~8-month lead-time, whereas a statistically significant relationship can only be simulated in about a quarter of the CMIP6 models (Fig. 2a). To determine the underlying mechanisms responsible for this, we rank the models based on their correlation between spring NTA SST anomaly and subsequent winter ENSO conditions, and then select the 10 models closest to the observations with the highest negative correlation (left side in Fig. 2a) and the 10 models most

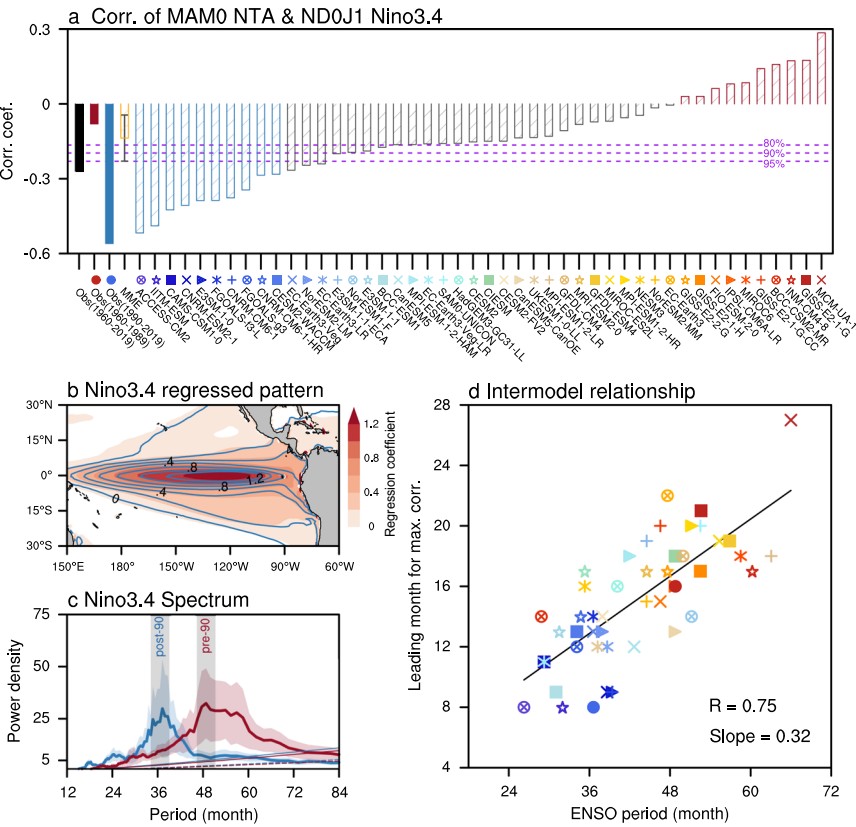

**Fig. 2 Phase relationship of NTA SST anomalies with ENSO in pi-control climate simulations. a** Lead correlation of boreal spring North Tropical Atlantic (NTA) sea surface temperature (SST) anomaly with the subsequent winter Niño3.4 index for 46 CMIP6 models and observations as a reference. The models are ranked by the NTA/ENSO correlation coefficients in an ascending order. The error bar for the multi-model ensemble (MME) mean corresponds to one standard deviation. The dashed purple lines represent the 80, 90, and 95% confidence levels. **b** Regression of SST anomalies (°C) upon the Niño3.4 index averaged for the left 10 models with most negative correlation (contours with interval: 0.4 °C; models indicated by striped blue bars in panel **a** and the right 10 models with most positive correlation (shading; models indicated by striped red bars in panel **a**). **c** Multi-Taper-Method (MTM) power spectra averaged for the left 10 models with most negative correlation (solid thick blue) and the right 10 models with most positive correlation (solid thick red), superimposed by one standard deviation (blue and red shading). The observed spectral peaks of pre- and post-1990 periods (gray shading) are shown for comparison. The averaged AR(1) null hypothesis is displayed by a dashed thin line and the 95% confidence level is indicated by a solid thin line. **d** Scatterplot of ENSO period and lead-time for which negative correlation coefficients are maximized for boreal spring NTA SST anomaly with the subsequent Niño3.4 index. The linear fit (solid black) is displayed together with the correlation coefficient R and slope.

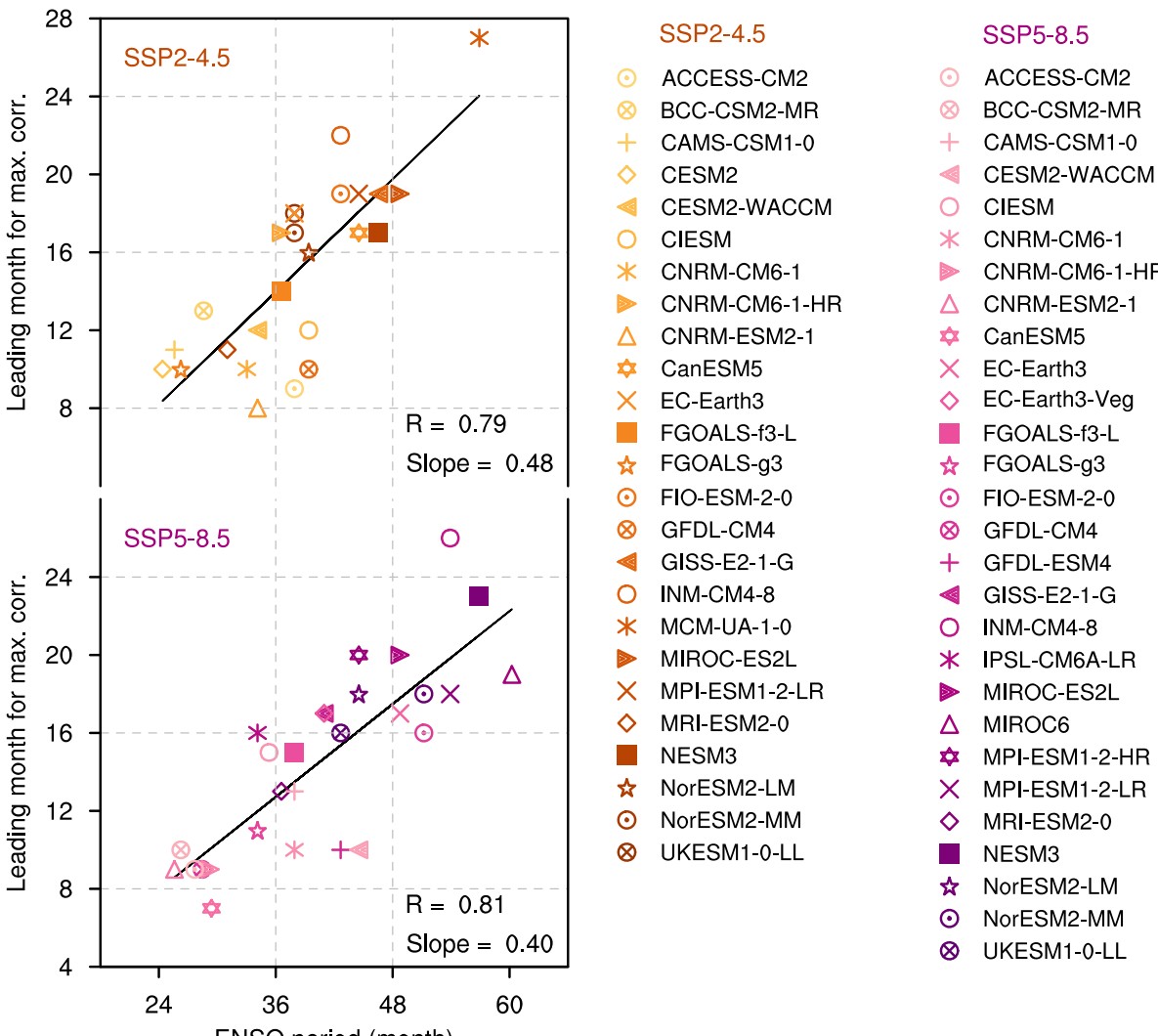

**Fig. 3 Phase relationship of NTA SST anomalies with ENSO in future warming simulations.** Scatterplot of ENSO period and lead-time at which negative correlation coefficients are maximized for boreal spring North Tropical Atlantic sea surface temperature anomaly with the subsequent Niño3.4 index for different climate models (different markers) in Shared Socioeconomic Pathways (SSP) 2-4.5 (red) and SSP5-8.5 (purple) scenarios. The linear fits (solid black) are displayed together with respective correlation coefficient $R$ and slope.

different from the observations that show a weakly positive correlation (right side in Fig. 2a).

Although both model groups show a very similar ENSO SST anomaly pattern (Fig. 2b), these two groups exhibit distinct ENSO spectral characteristics (Fig. 2c). The models that have a statistically significant 8-month ENSO-NTA lagged relationship exhibit a relatively shorter ENSO periodicity, analogous to the observations after the 1990s (Fig. 2c). In contrast, the models without a significant relationship at 8-month NTA-lead-time have longer ENSO periodicities resembling the observations before the 1990s (Fig. 2c). In addition, there is a high inter-model linear correlation ($R = 0.75$, statistically significant at the 95% confidence level) between simulated dominant ENSO periodicity and the lead-time of the most pronounced negative correlation of NTA SST leading ENSO (Fig. 2d). This again supports our hypothesis that the statistical lead-time of NTA SST anomalies over the subsequent ENSO conditions is tightly controlled by the ENSO periodicity.

There exists considerable uncertainty in the projections of trans-basin interactions and the pan-tropical climate patterns that will emerge in a warming world[23]. Thus, we next investigate the ENSO-NTA trans-basin interaction in CMIP6 future greenhouse-gas emission scenarios (see Methods). We find that almost all of these models in the SSP2-4.5 (25 of 25) and SSP5-8.5 (26 of 28) simulations are able to simulate the robust ENSO effect on the subsequent spring NTA SST (Supplementary Fig. 7). In turn, the linear relationship between NTA-lead-time over ENSO and ENSO periodicity continues to hold in the global warming scenarios (Fig. 3). High correlations can be detected in both warming scenarios ($R = 0.79$ for the SSP2-4.5 scenario and $R = 0.81$ for the SSP5-8.5 scenario, exceeding 95% confidence level). It further supports that the trans-basin ENSO-NTA relationships are predominantly determined by ENSO and its internal pacing.

## Discussion

In summary, ENSO plays a leading role in generating NTA SST variability in boreal spring following its peak phase via seasonally modulated atmospheric forcing and further influenced by the local SST adjustment timescale in the Atlantic (upper-left quadrant in Fig. 4). In turn, the observed time-varying relationship between these ENSO-induced NTA SST anomalies and the following ENSO conditions (Fig. 1c) can be explained by the ENSO regime shifting from dominantly quasi-quadrennial to dominantly quasi-biennial

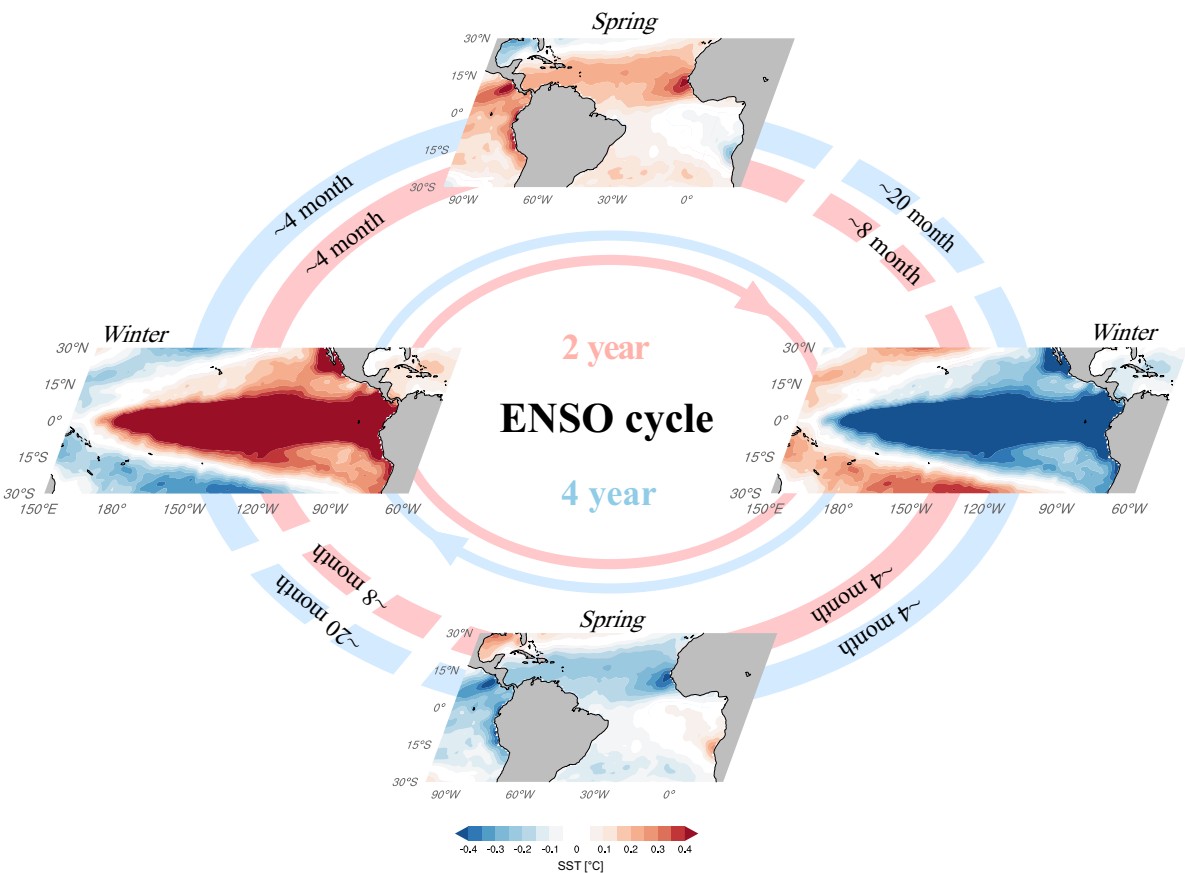

**Fig. 4 Schematic trans-basin relationships between tropical Pacific and North Atlantic oceans regulated by the ENSO periodicity.** In the quasi-biennial ENSO cycle (red loop), an El Niño condition in boreal winter (left panel) leads to positive North Tropical Atlantic (NTA) warming during subsequent spring (upper panel) at a ~4-month lead time, which in turn can see a La Niña formation (right panel) typically following El Niño in the subsequent winter, showing a statistical ~8-month lead time of the NTA. Likewise, a La Niña condition in boreal winter (right panel) gives rise to the following spring NTA sea surface temperature (SST) cooling (lower panel) with a lag of ~ 4 months, which is often followed by an El Niño formation (left panel), corresponding to a statistical ~8-month lead time of the NTA. The same applies for the quasi-quadrennial ENSO cycle (blue loop) except for the negative correlation of NTA SST variability with the following ENSO event by ~20 months.

around the 1990s (upper-right quadrant in Fig. 4). We emphasize that the observed ENSO cycles are not perfect oscillations with single frequencies. In nature, stochastic noise and nonlinearities can play important roles in shaping ENSO characteristics[41].

Here we demonstrated that the character of the observed cross correlation between ENSO and NTA is consistent with the hypothesis of an ENSO forced system. Nevertheless, does the identified ENSO-regime dependent NTA-ENSO relationship in-turn add any additional information useful for ENSO prediction, if the dominant ENSO regime (quasi-quadrennial or quasi-biennial) could be determined in advance? We take the quasi-biennial regime after the 1990s as an example considering less disturbance from the stochastic forcing in the relatively short time evolution of ENSO transition compared to the quasi-quadrennial regime before the 1990s. ENSO events with appreciable 8-month leading NTA signals (Supplementary Fig. 8) have in fact opposite-signed ENSO-related SST anomaly conditions in the previous boreal winter (Supplementary Figs. 9 and 10). For other ENSO years, no evident NTA-ENSO relationship can be detected. The notion of NTA serving as precursor for ENSO is therefore equivalent to simply saying that an El Niño is precursor to the next La Niña (Supplementary Fig. 10).

We conclude that previous suggestions about a possible NTA precursor leading to improved ENSO predictability and capacitor

arguments remain spurious based on observational data, a conceptual seasonally modulated ENSO-forced model, idealized pacemaker experiments, and CMIP6 simulations. We further show that our main results are robust even in a warming world.

## Methods

**Observation and statistics.** The utilized SST datasets are the global sea ice and SST analyses (1960–2019) from the Hadley Centre (HadISST) provided by the Met Office Hadley Centre with the horizonal resolution of 1° longitude × 1° latitude[42]. Anomalies were derived relative to the monthly mean climatology over the entire study period (1960–2019). A linear trend was removed to avoid possible influences associated with global warming. The Multi-Taper method (MTM), which uses a median smoother to distinguish signals from background noises, is used for spectral estimates[43] with 3 (Supplementary Fig. 1b) or 5 tapers (Figs. 2 and 3 and Supplementary Fig. 3) in consideration of different sample sizes. We test the spectra against the null hypothesis of an autoregressive model of order one (AR(1)) and calculate the respective 95% confidence levels. A nine-point smoothing is applied in Fig. 1c–e to avoid possible noise disturbance. All statistical significance tests were performed using the two-tailed Student's t test. El Niño events were identified according to the definition of the Climate Prediction Center based on a threshold of ±0.5 °C of the Niño3.4 index (averaged SST anomaly in the domain of 5°S–5°N, 120°–170°W) for five consecutive months. EP and CP indices (EPI and CPI) are calculated using a mathematic rotation of the Niño3 (averaged SST anomaly in the domain of 5°S to 5°N, 90° to 150°W) and Niño4 (averaged SST anomaly in the domain of 5°S to 5°N, 160°E to 150°W) indices[44]. El Niño events with EPI greater than CPI were classified as EP events while those with CPI greater than EPI are defined as CP events. Following this criterium, we identified seven EP El Niño events (1965, 1972, 1976, 1982, 1991, 1997, 2015) and thirteen CP El Niño

events (1963, 1968, 1969, 1977, 1979, 1986, 1994, 2002, 2004, 2006, 2009, 2018, 2019). The ENSO events during which the hindcasted Niño3.4 amplitudes attain more than 20% of the observed amplitudes due to previous spring NTA SST anomalies are defined as ENSO events with appreciable NTA contributions (Supplementary Fig. 7). The results are not sensitive to the criterion of the percentage chosen.

**Conceptual physical model.** We proposed a physically motivated model for the NTA SST anomaly as an extension[21] of the stochastic climate model:[40]

$$\frac{dT(t)}{dt} = \left(-\lambda_0 + \lambda_a\cos\left(\omega_a + \varphi\right)\right)T(t) + \beta ENSO(t) + \xi(t), \quad (1)$$

where $T(t)$ is the monthly NTA SST anomaly, ENSO(t) the monthly Niño 3.4 index, $\left(-\lambda_0 + \lambda_a\cos\left(\omega_a + \varphi\right)\right)$ the seasonally modulated damping rate, in which $\lambda_0$ and $\lambda_a$ denote the mean and annul cycle of the damping coefficient, $\omega_a$ the frequency of the annual cycle, $\varphi$ the phase shift, and $\beta$ a scaling coefficient. The model parameters are estimated by multivariate linear regression using the observed NTA SST anomaly time series and Niño 3.4 index following ref. [45] (Supplementary Table 1). The ENSO-independent stochastic forcing term $(\xi(t))$ is neglected in the model for simplicity S.

**CMIP6 simulations.** Monthly SST outputs from the CMIP6 pi-control and future (the Shared Socioeconomic Pathways (SSP) 2-4.5 and SSP5-8.5) simulations are utilized. The external forcing (e.g., greenhouse gases and aerosols) is kept constant in the pi-control simulations while the SSP2-4.5 with radiative forcing reaching 4.5 W m$^{-2}$ and SSP5-8.5 reaching 8.5 W m$^{-2}$ during 2015–2100[46,47]. For the pi-control simulations, the last 100 years of 46 available model simulations are used for the analysis, among which 25 models are obtained for the SSP2-4.5 scenario and 28 models for the SSP5-8.5 scenario, respectively (see Supplementary Table 2). Only one ensemble member for each model is used, mostly r1i1p1f1 with select models using ensemble member f2.

**Idealized pacemaker experiments.** Numerical experiments are conducted by using the Geophysical Fluid Dynamics Laboratory coupled model, version 2.1 (GFDL-CM2.1), with a horizontal resolution of 2.5° longitude × 2° latitude and 24 vertical levels[48]. Four sensitivity experiments are performed by using an idealized sinusoidal EP and CP ENSO forcing with 2- and 4-yr periodicities, respectively. Composited EP El Niño SST anomalies over the tropical Pacific (25°S–25°N, 150° E–90°W) are used to derive the SST anomalies forcing patterns for the EXP_2yr_EP experiment with repeated sinusoidal 2-yr periodicity and the EXP_4yr_EP experiment with repeated 4-yr periodicity. The other two experiments (EXP_2yr_CP and EXP_4yr_CP) are the same, except that the SST anomalies are the composites for the observed CP El Niño events. SST anomalies outside the forcing area are set to zero and only the positive loading in the forcing region is used. These SST anomaly patterns are then added to the 1969–2019 climatological SSTs. The SSTs are allowed to evolve freely outside of the prescribed regions. ENSO peak phases will occur aligned with the boreal winter season for these idealized 2- and 4-yr periodicities. The simulations are integrated for 100 years and the output from the last 80 years is used for the analyses. Anomalies in GFDL-CM2.1 are relative to a 100-year control simulation (EXP_CTRL) in which the model is forced with seasonal varying climatological SSTs.

## Data availability

The data used to reproduce the results of this paper are available online or by contacting the corresponding author. Hadley SST data is publicly available at: https://www.metoffice.gov.uk/hadobs/hadisst/data/download.html. The CMIP6 datasets are available at https://esgf-node.llnl.gov/projects/cmip6/.

## Code availability

Codes used in this study are available from the corresponding author on request.

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

## Acknowledgements
This work was supported by the National Nature Science Foundation of China (42088101) and the National Key Research and Development Program (2018YFC1506002). M.F.S. was supported by NOAA's Climate Program Office's Modeling, Analysis, Predictions, and Projections (MAPP) program grant NA20OAR4310445. This is IPRC publication 1518 and SOEST contribution 11327. A.T. was supported by the Institute for Basic Science (project code IBS-R028-D1).

## Author contributions
W.Z., F.J., M.F.S., F.F.J., and A.T. conceived the idea. W.Z. and F.J. conducted the data analyses and prepared the figures. W.Z., F.J., M.F.S., F.F.J., and A. T. discussed the results and wrote the paper.

## Competing interests
The authors declare no competing interests.
