## [Peer Review File · Nature Communications]

Reviewers' Comments:

Reviewer #1:

Remarks to the Author:

I have reviewed the manuscript 'Spurious North Tropical Atlantic pre-cursors to ENSO', by W. Zhang et al.

This is an interesting and well researched study. It clearly shows that the teleconnection from ENSO to the tropical North Atlantic alone may be able to explain observed lead-lag correlations between them, with out a feedback from the North Atlantic to ENSO being necessary. As such this study claims to provide evidence that a previous study by Ham et. al., also in a Nature publishing group, is not valid. It is curious that the Ham paper and this manuscript share a co-author.

Anyway, I do think that the current study actually does not disprove the findings from Ham et. al., and therefore the language of this paper (including probably title) should be considerably turned down. I recommend a major revision addressing the issues below:

1. The fact that this manuscript shows that ENSO's teleconnection to the North Atlantic can explain also their lead-lag correlations does not mean that they have shown that a feedback from the North Atlantic to ENSO is also relevant and may also contribute the the lead-lag correlations. This is basic logic.
2. To further support point one, what this manuscript does is essentially only provides evidence for the one-way influence of ENSO on the North Atlantic. However, note that Ham et al., do exactly the opposite, so I guess their results may be complementary. Note that Ham et al, as well as other studies (e.g. Kucharsi et. al., Atmosphere 2016, 7(2), 29; <https://doi.org/10.3390/atmos7020029>) remove the previous winter ENSO statistically to show the feedback on the following year ENSO. You may repeat some of your analysis also removing the previous year's ENSO in order to check the impact of this on your results. Perhaps we see that some of the lead-lag relation could be indeed related to the feedback on the tropical North Atlantic on ENSO.
3. Furfurthermore, the 2 studies mentioned in point 2 also provide pacemaker experiments to show the feedback from the tropical North Atlantic to ENSO, just the opposite of what the present manuscript does. Therefore, these studies are really complementary.
4. In your idealised experiments (e.g. lines 130 - 150), you prescribe a perfectly autocorrelated ENSO cycle. Then your results are almost granted. The correlation for El Nino to the following La Nina or vice versa is in reality much weaker, and the feedback from the North Atlantic may therefore in reality be important to explain the ENSO periodicity.
5. The use of CMIP models gives in no way more confidence tothe main results that the ENSO-North tropical Atlantic interaction is in reality one-way. It has been well noted in many studies that ENSO's impact may be overestimated in many cmip models, whereas the impact of other ocean basins on ENSO could therefore be underestimated.

Reviewer #2:

Remarks to the Author:

Review of "Spurious North Tropical Atlantic pre-cursors to ENSO"

The paper by Wenjun Zhang and his co-authors revisits the linear two-way relationship between ENSO and the Tropical North Atlantic (TNA) sea surface temperature (SST) variability by taking into consideration important aspects of ENSO dynamics, e.g. its strong autocorrelation and irregular cycle. Previous studies have shown that ENSO has a significant impact on the TNA with a lag of 4 months (consistent with the SSTs delayed response). Previous studies suggest that spring TNA SST anomalies

can be a pre-cursor to ENSO in the next season with a lag of 8 months, although this correlation is only present after the 1990s. Before the 1990s the TNA lags the ENSO response in the Pacific by approximately 20 months, which would be consistent with a quasi-quadrennial cycle, in contrast with the quasi-biennial cycle dominant in the more recent period. The authors use a rich variety of tools (observations, a simple physical model, output from CIMP6 models, and idealized pacemaker experiments) to prove that the suggested impact of the TNA variability on ENSO development is negligible if we account for the internal ENSO cycle. The authors also prove that these conclusions are robust under future climate scenarios.

The results presented in this manuscript are very interesting and relevant for a wide scientific audience. I really enjoyed reading this paper. It is well-written and presents a clear message, which is proved in a very systematic and convincing way. I highly recommend this paper for publication in Nature communications. I just have some small minor comments, which are mostly some clarifications on the methods employed.

Minor comments:

Lines 62-64: The authors describe the mechanism for the warm phase of ENSO. Does La Niña also lead to a similar response on the TNA? From a correlation map, it is difficult to say whether this is the case.

Lines 96-97: How do the authors justify that the complicated irregular ENSO cycle does not affect this relationship? A better reason/explain would be desired.

Line 99-101: I do not understand why a bandpass of 2-3yr is applied to extract the quasi-biennial periodicity and a 3-5yr is used to extract the quasi-quadrennial cycle. Should not they be consistent, i.e. 1-3yr and 3-5yr? I guess this shouldn't affect the results of the study, but it will be interesting to justify the 2-3yr choice instead of 1-3yr. Is it to remove the seasonal cycle?

Lines 145-146: I think this is a mistake. The quasi-biennial ENSO cycle and 8 months lag between the TNA and the following ENSO phase is after the 1990s.

Lines 161-162: Looking at supplementary Figure 5 it seems that there is a large inter-model spread. It is however impressive how the multi-model mean captures the ENSO-TNA correlation so well. I think the model spread should be mentioned here.

Line 190: I do not understand the term "In-turn linear relationship". Maybe rewrite this sentence.

Lines 220-221: Do the analysis change when not removing a linear trend? The authors show later that this relationship is still valid even under climate change conditions.

Lines 246-248: What are the parameters that the authors obtain after using the observations to fit this simple model.

Line 248-249: What would be the effect of including this term? What is the reason to not include it?

Line 252: What is the difference between "SSP2-4.5 and SSP5-8.5" and "RCP4.5 and RCP8.5" climate emission scenarios? I think the climate community is more familiar with the RCP scenarios.

Line 277: What SST climatology is used (observations or model, period)?

Figure 1e: I cannot see which lines are dashed and which ones are not in the legend (CP vs EP). This is also not specified in the figure caption.

Sincerely,
Bernat Jiménez-Esteve

Response to Reviewers

We would like to thank the reviewers for their constructive comments and suggestions. We have addressed the reviewers' concerns and implemented the useful suggestions in our revised manuscript. The responses are listed below in blue font.

Response to Reviewer#1

I have reviewed the manuscript 'Spurious North Tropical Atlantic pre-cursors to ENSO', by W. Zhang et al.

This is an interesting and well researched study. It clearly shows that the teleconnection from ENSO to the tropical North Atlantic alone may be able to explain observed lead-lag correlations between them, without a feedback from the North Atlantic to ENSO being necessary. As such this study claims to provide evidence that a previous study by Ham et. al., also in a Nature publishing group, is not valid. It is curious that the Ham paper and this manuscript share a co-author.

Anyway, I do think that the current study actually does not disprove the findings from Ham et. al., and therefore the language of this paper (including probably title) should be considerably turned down. I recommend a major revision addressing the issues below:

Response: Thank you for your valuable comments. In this manuscript, we demonstrate that the cross-correlation characteristics between NTA SST and ENSO are consistent with a one-way Pacific to Atlantic forcing. Our work here identified the appropriate way to formulate a physical null hypothesis – that is, by considering the different ENSO regimes – that would need to be falsified by future studies that claim a two-way feedback.

In addition, we present multiple lines of evidence that the NTA SST variability cannot be identified as a statistically significant pre-cursor for ENSO, regardless of whether a feedback from the North Atlantic on ENSO exists based on both observations and a physical model in the manuscript and the following point-to-point responses. Furthermore, we also demonstrate that the results from the previous NTA forced pacemaker experiments by Ham et al. (2013a,b) cannot be used as evidence to reject our null hypothesis of a one-way forcing.

As mentioned by the reviewer, Ham et al. (2013a) and this manuscript shared one co-author (Prof. Fei-fei Jin), who is also one of the corresponding authors here. The concept that the NTA variability has a possible feedback on the following ENSO was first advanced in Ham et al. (2013a). This study was then followed by Wang et al. (2017), which deepened our understanding that this relationship is statistically nonstationary. They attempted to attribute the nonstationarity to the AMO phase transition from negative to positive around the 1990s. However, this hypothesis cannot explain why the correlation between spring NTA SST and following ENSO is also statistically insignificant during the previous positive AMO period (before the

1960s). Apparently, the previous argument that global warming together with the positive phase of the AMO contribute to the relationship is without substance.

This inspired us (including Prof. Fei-Fei Jin) to revisit the intriguing question. Here, we find that the relationship between NTA SST and ENSO is consistent with a one-way Pacific to Atlantic forcing null hypothesis. This framing by ENSO one-way forcing explains the observed NTA-ENSO linkage variations over the whole observational period in a consistent way. Moreover, the regulation of the ENSO cycle on the NTA-ENSO relationship continues to hold in future warming scenarios, which again disproves the previous assumption on this relationship.

1. The fact that this manuscript shows that ENSO's teleconnection to the North Atlantic can explain also their lead-lag correlations does not mean that they have shown that a feedback from the North Atlantic to ENSO is also relevant and may also contribute the the lead-lag correlations. This is basic logic.

Response: Thank you for your valuable comment. In this manuscript we would like to clarify that the NTA SST cannot be statistically identified as a pre-cursor for ENSO regardless of whether a feedback from the North Atlantic on ENSO exists. This is a crucial point that is of importance to future studies investigating any potential inter-basin feedbacks.

As we have shown in Supplementary Fig. 3 in the manuscript, the residual NTA SST variability – once the ENSO signal is removed – has no preferred interannual spectral peak. This strongly suggests that NTA SST interannual variability could originate from ENSO. In the manuscript, we only consider a simplified ENSO-forced NTA model in which the NTA is treated as a purely deterministic process. From a possible realistic perspective, we could include the stochastic forcing (Eq. (1)) in our original ENSO-forced model (Eq. (2)) (Jin et al. 2007),

$$\frac{d\xi(t)}{dt} = -m\xi(t) + w(t), \quad (1)$$

$$\frac{dT(t)}{dt} = (-\lambda_0 + \lambda_a \cos(\omega_a + \varphi))T(t) + \beta ENSO(t) + \xi(t), \quad (2)$$

where $w(t)$ indicates white noise with a Gaussian distribution and the decorrelation time scale parameter m is set to 1/2 month here. An ensemble of 500 members of red noise times series ξ was generated. We integrated Eq. (2) for 60 years to create a 500-member ensemble. The observed NTA SST variability can be sufficiently explained by the reconstructed 500-member ensemble mean NTA with ± 0.5 standard deviation spread (Fig. R1). This indicates that the NTA SST variability is fully consistent with an ENSO-forced stochastic dynamical process.

However, does this forced NTA variability in-turn account for any ENSO predictability? This is the major concern from the reviewer and our argument here is that the NTA variability actually adds no additional information in ENSO prediction. We demonstrate this argument from two aspects.

(1) Observational evidence

The 8-month leading relationship of NTA over ENSO only exists within the

chain of the ENSO cycle (i.e., ENSO transition from El Niño to La Niña or vice versa) (Fig. R2-R5). To quantify the possible NTA feedback on ENSO, we compare the observed and hindcasted Niño3.4 index by using the previous spring NTA SST. The El Niño and La Niña events in which the hindcasted Niño3.4 amplitudes attain more than 20% of the observed amplitudes are defined as ENSO events with a positive NTA contribution. The results are not sensitive to the criterion of the percentage chosen.

All ENSO years with positive NTA contributions (1970, 1976, 1983, 1986, 1994, 1998, 2005, 2009, 2010, 2016, 2018) have in fact opposite ENSO-related SST anomaly conditions in the previous boreal winter (Fig. R3 and Fig. R4). In other words, the so-called contribution from NTA originates entirely from the previous ENSO conditions. For these ENSO years with positive NTA contributions, we can also hindcast the Niño3.4 index by using the previous winter Niño3.4 index (Red dots in Fig. R5b). In particular, there even appears a significant positive correlation between the previous spring NTA SST and winter ENSO for the second/third year of consecutive ENSO events (1969, 1971, 1974, 1975, 1977, 1984, 1987, 1999, 2000, 2008, 2011, 2017, 2019). The opposite relationship between NTA SST and following ENSO seen in those ENSO years can also be explained by the correlation between the winter Niño3.4 index with previous winter Niño3.4 index (Blue dots in Fig. R5). Importantly, for other ENSO years no evident NTA-ENSO relationship can be detected. This again supports our argument in the manuscript “*The notion of NTA serving as precursor for ENSO is therefore equivalent to simply saying that the El Niño is precursor to the next La Niña.*”

(2) Simulation insufficiencies

a) The pacemaker experiments in Ham et al. 2013b lack the key pathway for NTA to impact ENSO, in which the observed SST anomalies in the eastern subtropical and tropical North Pacific are absent (Fig. 1 versus Fig. 4 in Ham et al. 2013b).

b) The pacemaker experiments in Ham et al. 2013b fail to reproduce reasonable ENSO events with realistic phase locking features, which is a fundamental ENSO property. In their experiments, ENSO-related SST anomalies synchronize to the boreal autumn season (Sep-Oct-Nov) and quickly recede in the forthcoming winter (Fig. 4e,g in Ham et al. 2013b).

c) The ENSO-related essential atmospheric processes in those NTA-forced experiments show large model dependency. Strong westerly anomalies in the eastern subtropical and tropical North Pacific shown in Ham et al. 2013b, which is the key physical process, are missing in similar experiments conducted in one review paper mentioned by the reviewer (Kucharski et al. 2016).

Based on the above analyses, the NTA-ENSO relationship is tightly controlled by the ENSO cycle and the interannual NTA SST variability cannot feed back on ENSO in a predictable manner (see also in L15-19 in the manuscript). We have added some analyses and related discussion in the revised manuscript (L210-223). Besides, we also avoid using some absolutized expression to avoid possible confusion (e.g.,

L23 in the manuscript).

Figure R1. Time series of the observed monthly NTA (black line) and reconstructed monthly ensemble mean NTA (red line). Red shading denotes the ± 0.5 standard deviation spread within the 500-member ensemble.

Figure R2. Observed (red line) and hindcasted (black line) Niño3.4 index (°C) using the previous spring NTA SST. Light and dark grey rectangles indicate the El Niño and La Niña years with hindcasted Niño3.4 index accounting for more than 20% of the observation, respectively. The correlation coefficient between observed and hindcasted Niño3.4 index is 0.28.

er of a El Niño
20% amplitude.
nt at the 95%

Figure R4. a The I
for more than 20% ;
standard deviation.
NTA contribution (c
and gray bars the pr

Figure R5. Scatterplot of winter Niño3.4 index with **a** previous spring NTA SST anomaly and **b** previous winter Niño3.4 index. Red dots in **(a-b)** denote the ENSO years with NTA contribution accounting for more than 20% amplitude, blue dots the second/third years of consecutive ENSO events, orange dots the other ENSO years, and gray dots denote the residual years. The linear fits of red and blue dots are displayed together with the corresponding correlation coefficients (R). The correlation coefficient (R; black) for all dots is also shown.

2. To further support point one, what this manuscript does is essentially only provides evidence for the one-way influence of ENSO on the North Atlantic. However, note that Ham et al., do exactly the opposite, so I guess their results may be complementary. Note that Ham et al., as well as other studies (e.g. Kucharski et. al., Atmosphere 2016, 7(2), 29; <https://doi.org/10.3390/atmos7020029>) remove the previous winter ENSO statistically to show the feedback on the following year ENSO. You may repeat some of your analysis also removing the previous year's ENSO in order to check the impact of this on your results. Perhaps we see that some of the lead-lag relation could be indeed related to the feedback on the tropical North Atlantic on ENSO.

Response: Thank you for your comment. As we have discussed in our response to Comment#1, the ENSO-forced NTA variability does not account for additional ENSO predictability. Thus, our focus in this manuscript is that the NTA SST cannot be identified as a statistically significant pre-cursor for ENSO regardless of whether a feedback from the North Atlantic on ENSO exists.

It is somewhat accepted by default in many climate studies to exclude ENSO impacts by linearly removing the previous boreal winter ENSO signal, which is unreasonable considering that ENSO is a highly complicated and nonlinear climate phenomenon, considering its amplitude (the asymmetry of El Niño and La Niña episodes), temporal evolution (from weather, annual cycle, interannual to decadal timescales) and spatial pattern (eastern Pacific versus central Pacific El Niño)

Here, we extracted the ENSO signal from the NTA variability by using a simplified ENSO-forced NTA model (Supplementary Fig. 2). As shown in

Supplementary Fig. 3, the residual variability has no preferred interannual spectral peak.

We also show in our response to Comment#1 the observational evidence and the shortcomings of the previous NTA-forced pacemaker experiments.

3. Furthermore, the 2 studies mentioned in point 2 also provide pacemaker experiments to show the feedback from the tropical North Atlantic to ENSO, just the opposite of what the present manuscript does. Therefore, these studies are really complementary.

Response: Thank you very much for your comment. As shown in response to Comment#1, the NTA SST cannot be identified as the pre-cursor for ENSO regardless of whether a feedback from the North Atlantic on ENSO exists. In this manuscript, we are not intended to rule out the possible feedback of NTA on ENSO.

4. In your idealised experiments (e.g. lines 130 - 150), you prescribe a perfectly autocorrelated ENSO cycle. Then your results are almost granted. The correlation for El Niño to the following La Niña or vice versa is in reality much weaker, and the feedback from the North Atlantic may therefore in reality be important to explain the ENSO periodicity.

Response: Thank you very much for your comment. As shown in our response to Comment#1, the so-called NTA feedback only exists within the chain of the ENSO cycle (i.e., ENSO transition from El Niño to La Niña or vice versa). The NTA variability actually adds no additional information in ENSO prediction (Fig. R2-R5).

Besides, we have discussed in the manuscript (L97-98) that the complexity of the ENSO cycle does not affect the qualitative relationship of NTA SST with following ENSO from a statistical standpoint. This argument can be succinctly proved based on our NTA physical model. Three experiments are performed by using different ENSO cycles as in the case of a 4-yr cycle. A perfectly 4-yr sinusoidal ENSO cycle (similar to the experimental design of the pacemaker experiments in the manuscript) is prescribed in EXP1 for reference. Then a more complicated and realistic 4-yr ENSO cycle comprising 1-yr El Niño and 3-yr La Niña is prescribed in EXP2 (Fig. R6a and b). Besides, the possible interference from the amplitude asymmetry of El Niño and La Niña episodes is also examined in EXP3 by multiplying El Niño amplitude by a factor of 2 in EXP1. Using the parameters estimated in Supplementary table 1, we force the NTA model with different ENSO evolutions for 20 ENSO cycles (960 months) and then derive the respective NTA time series (Fig. R6c). Qualitative ENSO-NTA lead-lag relationships can be obtained under the consideration of these ENSO irregular behaviors including duration and amplitude asymmetries (Fig. R6d), despite slight differences in the respective correlation coefficients.

Figure R6. **a** ENSO time series ($^{\circ}\text{C}$) in EXP1 (black line), EXP2 (red line) and EXP3 (blue line) in two ENSO cycles. **b** Fast Fourier Transform (FFT) power spectra for ENSO time series. The AR(1) null hypothesis is displayed by a dashed thin line and the 95% confidence level is indicated by a solid thin line. **c** Derived NTA time series ($^{\circ}\text{C}$) in one ENSO cycle. **d** Lead-lagged correlation of the boreal winter ENSO with NTA time series.

5. *The use of CMIP models gives in no way more confidence to the main results that the ENSO-North tropical Atlantic interaction is in reality one-way. It has been well noted in many studies that ENSO's impact may be overestimated in many CMIP models, whereas the impact of other ocean basins on ENSO could therefore be underestimated.*

Response: Thank you very much for your valuable comment. We agree with your concern regarding biases between the multi-model estimation and the observed ENSO-NTA relationship. In Supplementary Fig. 5 we assess the ability of current climate models to simulate the ENSO-NTA connection. The multi-model ensemble mean and the majority of the individual models can reasonably reproduce the observed ENSO's impact on NTA SST, although with a certain inter-model spread. Thus, the analysis of the multi-model results again supports our hypothesis that the statistical lead-time of NTA SST anomalies over the subsequent ENSO conditions is tightly regulated by the ENSO periodicity. We have added some related discussion about the model performance in reproducing observed ENSO-NTA connection in the revised manuscript (L165-167).

Response to Reviewer#2

Review of “Spurious North Tropical Atlantic pre-cursors to ENSO”

The paper by Wenjun Zhang and his co-authors revisits the linear two-way relationship between ENSO and the Tropical North Atlantic (TNA) sea surface temperature (SST) variability by taking into consideration important aspects of ENSO dynamics, e.g. its strong autocorrelation and irregular cycle. Previous studies have shown that ENSO has a significant impact on the TNA with a lag of 4 months (consistent with the SSTs delayed response). Previous studies suggest that spring TNA SST anomalies can be a pre-cursor to ENSO in the next season with a lag of 8 months, although this correlation is only present after the 1990s. Before the 1990s the TNA lags the ENSO response in the Pacific by approximately 20 months, which would be consistent with a quasi-quadrennial cycle, in contrast with the quasi-biennial cycle dominant in the more recent period. The authors use a rich variety of tools (observations, a simple physical model, output from CIMP6 models, and idealized pacemaker experiments) to prove that the suggested impact of the TNA variability on ENSO development is negligible if we account for the internal ENSO cycle. The authors also prove that these conclusions are robust under future climate scenarios. The results presented in this manuscript are very interesting and relevant for a wide scientific audience. I really enjoyed reading this paper. It is well-written and presents a clear message, which is proved in a very systematic and convincing way. I highly recommend this paper for publication in Nature communications. I just have some small minor comments, which are mostly some clarifications on the methods employed.

Minor comments:

1. Lines 62-64: The authors describe the mechanism for the warm phase of ENSO. Does La Niña also lead to a similar response on the TNA? From a correlation map, it is difficult to say whether this is the case.

Response: Thank you for your valuable comment. La Niña events also lead to a similar response on the NTA SST variability (Fig. R7).

Figure R7. Cor
decaying spring.
significant at the

2. *Lines 96-97: F
does not affect th*

Response: The argument based on the simple model in the manuscript since experiments are performed by using different ENSO cycles as in the case of a 4-yr cycle. A perfectly 4-yr sinusoidal ENSO cycle (similar to the experimental design of the pacemaker experiments in the manuscript) is prescribed in EXP1 for reference. Then a more complicated and realistic 4-yr ENSO cycle comprising 1-yr El Niño and 3-yr La Niña is prescribed in EXP2 (Fig. R8a and b). Besides, the possible interference from the amplitude asymmetry of El Niño and La Niña episodes is also examined in EXP3 by multiplying El Niño amplitude by a factor of 2 in EXP1. Using the parameters estimated in Supplementary table 1, we force the NTA model with different ENSO evolutions for 20 ENSO cycles (960 months) and then derive the respective NTA time series (Fig. R8c). Qualitative ENSO-NTA lead-lag relationships can be obtained under the consideration of these ENSO irregular behaviors including duration and amplitude asymmetries (Fig. R8d), despite slight differences in the respective correlation coefficients. Above all, the complicated irregular ENSO cycle does not affect the relationship qualitatively. We have modified our description to make it clearer.

Figure R8. **a** ENSO time series ($^{\circ}\text{C}$) in EXP1 (black line), EXP2 (red line) and EXP3 (blue line) in two ENSO cycles. **b** Fast Fourier Transform (FFT) power spectra for ENSO time series. The AR(1) null hypothesis is displayed by a dashed thin line and the 95% confidence level is indicated by a solid thin line. **c** Derived NTA time series ($^{\circ}\text{C}$) in one ENSO cycle. **d** Lead-lagged correlation of the boreal winter ENSO with NTA time series.

3. Line 99-101: I do not understand why a bandpass of 2-3yr is applied to extract the quasi-biennial periodicity and a 3-5yr is used to extract the quasi-quadrennial cycle. Should not they be consistent, i.e. 1-3yr and 3-5yr? I guess this shouldn't affect the results of the study, but it will be interesting to justify the 2-3yr choice instead of 1-3yr. Is it to remove the seasonal cycle?

Response: In the manuscript, we apply a bandpass of 2-3yr to avoid possible affects by of near-annual variability, such as the ENSO combination mode (C-mode) (Stuecker et al. 2013). Actually, it does not affect our conclusion and qualitative results can be obtained by applying 1-3yr bandpass filtering (Fig. R9).

with the filtering period. For a lag at

7 and 8

a large increase there.

the mean observed. We have the manuscript

Maybe

rewrite this sentence.

Response: Thank you for your valuable suggestion. We have modified our related discussion as “In turn, the linear relationship between NTA-lead-time over ENSO and ENSO periodicity continues to hold in the global warming scenarios.”

7. Lines 220-221: Do the analysis change when not removing a linear trend? The authors show later that this relationship is still valid even under climate change conditions.

Response: Thank you for your valuable comment. The analysis does not change

V

I

ξ

c

I

e

ξ

t

I

consider a simplified ENSO-forced NTA model. As we have shown in Supplementary Fig. 3, the residual NTA SST variability – once the ENSO signal is removed – has no preferred interannual spectral peak. This strongly suggests that NTA SST interannual variability could originate from ENSO. In the manuscript, we only consider a simplified ENSO-forced NTA model in which the NTA is treated as a purely deterministic process. From a possible realistic perspective, we could include the stochastic forcing (Eq. (1)) in our original ENSO-forced model (Eq. (2)) (Jin et al. 2007),

$$\frac{d\xi(t)}{dt} = -m\xi(t) + w(t), \quad (3)$$

$$\frac{dT(t)}{dt} = (-\lambda_0 + \lambda_a \cos(\omega_a + \varphi))T(t) + \beta ENSO(t) + \xi(t), \quad (4)$$

where $w(t)$ denotes white noise with a Gaussian distribution and the decorrelation time scale parameter m is set to 0.5 month here. An ensemble of 500 members of red noise times series ξ was then generated. We integrated the Eq. (4) for 60 years to create a 500-member ensemble. The observed NTA SST variability can be sufficiently explained by the reconstructed 500-member ensemble mean NTA with ± 0.5 standard deviation spread (Fig. R11). This also indicates that the NTA SST variability is fully

consistent with an ENSO-forced stochastic dynamical process. In this manuscript, we neglect this term to emphasize the importance of ENSO forcing on the NTA SST, considering that the predictable part of the NTA SST variability mainly originates from ENSO.

Figure R11. Time series of the observed monthly NTA (black line) and reconstructed monthly ensemble mean NTA (red line). Red shading denotes the ± 0.5 standard deviation spread within the 500-member ensemble.

10. Line 252: What is the difference between “SSP2-4.5 and SSP5-8.5” and “RCP4.5 and RCP8.5” climate emission scenarios? I think the climate community is more familiar with the RCP scenarios.

Response: Thank you for your comment. Future simulations in CMIP5 were made under a set of four pathways of anthropogenic greenhouse gas and aerosol concentrations resulting in different levels of radiative forcing—the Representative Concentration Pathways (RCPs; Moss et al. 2010; Vuuren et al. 2011). The new scenarios in CMIP6 combine a RCP and a Shared Socioeconomic Pathway (SSP; O’Neill et al. 2016), named for the SSP value then the RCP value. The RCPs, which are defined by the magnitude of radiative forcing at 2100, contain the same values as for CMIP5 and also some new ones: 1.9, 2.6, 3.4, 4.5, 6.0, 7.0, and 8.5 W/m^2 . The five SSPs are summarized by the narrative headlines; SSP1-sustainability: taking the green road; SSP2-middle of the road; SSP3-regional rivalry: a rocky road; SSP4-inequality: a road divided; and SSP5-fossil-fueled development: taking the highway (O’Neill et al. 2016). The framing in CMIP6 including both SSP and RCP dimensions offers opportunities to examine the future in terms of physical climate change, socioeconomic pathways and their possible interactions.

11. Line 277: What SST climatology is used (observations or model, period)?

Response: Thank you for the comment. The SST anomalies derived from observations are added to the 1960-2019 climatological SSTs. We have added related description in the revised manuscript (L290-291) to make this issue clear.

12. Figure 1e: I cannot see which lines are dashed and which ones are not in the

legend (CP vs EP). This is also not specified in the figure caption.

Response: Thank you for your suggestion. We have improved Figure 1 for better elucidation. The details of the experimental design are shown in the Methods, so we do not repeat this information in the caption for simplicity.

References

- Kucharski, F. et al. The Teleconnection of the Tropical Atlantic to Indo-Pacific Sea Surface Temperatures on Inter-Annual to Centennial Time Scales: A Review of Recent Findings. *Atmosphere* 7(2), 29 (2016).
- Jin, F.-F., Lin, L., Timmermann, A. & Zhao, J. Ensemble-mean dynamics of the ENSO recharge oscillator under state-dependent stochastic forcing. *Geophys. Res. Lett.* 34, L03807 (2007).
- Ham Y.-G., Kug J.-S., Park J.-Y. & Jin F.-F. Sea surface temperature in the north tropical Atlantic as a trigger for El Niño/Southern Oscillation events. *Nature Geosci.* 6, 112–116 (2013a).
- Ham Y.-G., Kug J.-S. & Park J.-Y. Two distinct roles of Atlantic SSTs in ENSO variability: North tropical Atlantic SST and Atlantic Niño. *Geophys. Res. Lett.* 40, 4012–4017 (2013b).
- Moss, R. H. et al. The next generation of scenarios for climate change research and assessment. *Nature* 463, 747–756 (2010).
- O’Neill, B. C. et al. The scenario model intercomparison project (ScenarioMIP) for CMIP6. *Geosci. Model Dev.* 9, 3461–3482 (2016).
- Stuecker, M. F., Timmermann, A., Jin, F.-F., McGregor, S. & Ren, H.-L. A combination mode of the annual cycle and the El Niño/Southern Oscillation. *Nat. Geosci.* 6, 540–544 (2013).
- van Vuuren, D. et al. The representative concentration pathways: An overview. *Climatic Change.* 109(1–2), 5–31 (2011).
- Wang L., Yu J.-Y. & Paek H. Enhanced biennial variability in the Pacific due to Atlantic capacitor effect. *Nature Commun.* 8, 14887 (2017).

Reviewers' Comments:

Reviewer #1:

Remarks to the Author:

The authors have addressed all my concerns. Therefore the paper may be accepted.

Reviewer #2:

Remarks to the Author:

The revised version of the manuscript by W. Zhang et al. is well written and the results concerning the ENSO-NTA relationship using different methodologies remain the same as the previous version. In general, although the authors have not made any substantial changes in the manuscript, they have put enough effort into answering and clarifying most of my previous comments, as well as the more critical ones by the other reviewer. From my side, I am satisfied with the answers and justifications given to my comments, but I would have appreciated that more of these comments would have been introduced in the manuscript. It would have also helped to have a version with tracked changes, otherwise, I assume not much has been changed. I just have some more specific very minor comments on the text. Apart from those, I am happy to recommend this paper for publication in Nature Communications.

Minor comments:

Line 55: Maybe rephrase the sentence. In my opinion, in this paper, you are not analyzing the underlying mechanisms, but the statistical robustness of the time-varying relationship. You propose a mechanism, but you don't analyze it dynamically.

Line 64-65: Specify how many months.

Line 75: you might remove "even"

Lines 97-99: I think referring to Figure R6 here, and add it to the supplementary material, would support this claim, which is not obvious that it is true.

Lines 150-155: When comparing in more detail figures 1d and 1e, one can see that whereas the correlation for the 2yr-cycle matches quite well with the 2-3yr filtered observations, the 4yr-cycle experiments show a slightly different evolution than the 3-5yr filtered correlation. This is not mentioned in the manuscript. I wonder if this is due to the simulation setup.

Sincerely,

Bernat Jiménez-Esteve

Response to Reviewers

We would like to thank the reviewers for their constructive comments and suggestions. We have addressed the reviewers' concerns and implemented the useful suggestions in our revised manuscript. The responses are listed below in blue font.

Response to Reviewer#2

The revised version of the manuscript by W. Zhang et al. is well written and the results concerning the ENSO-NTA relationship using different methodologies remain the same as the previous version. In general, although the authors have not made any substantial changes in the manuscript, they have put enough effort into answering and clarifying most of my previous comments, as well as the more critical ones by the other reviewer. From my side, I am satisfied with the answers and justifications given to my comments, but I would have appreciated that more of these comments would have been introduced in the manuscript. It would have also helped to have a version with tracked changes, otherwise, I assume not much has been changed. I just have some more specific very minor comments on the text. Apart from those, I am happy to recommend this paper for publication in Nature Communications.

Minor comments:

Line 55: Maybe rephrase the sentence. In my opinion, in this paper, you are not analyzing the underlying mechanisms, but the statistical robustness of the time-varying relationship. You propose a mechanism, but you don't analyze it dynamically.

Response: Thank you for your valuable suggestion. We have rephrased the sentence in L57 in the revised manuscript as “In this study, we use both observations and climate model simulations to interpret this time-varying relationship.”

Line 64-65: Specify how many months.

Response: Thank you for your valuable suggestion. We have specified the months in L67 in the revised manuscript as “Analyzing observed SST anomalies (see Methods), we see that the El Niño remote forcing is felt in the NTA SST 3-5 months later around the spring season.”

Line 75: you might remove “even”

Response: Thank you for your valuable suggestion. We have removed “even” in the revised manuscript (L78).

Lines 97-99: I think referring to Figure R6 here, and add it to the supplementary

material, would support this claim, which is not obvious that it is true.

Response: Thank you for your valuable suggestion. We have added Figure R6 and some discussion in the supplementary information. Considering that we have not introduced the ENSO-forced NTA model here (L97-99 in the last version), we move the related discussion to L125-130 in the revised manuscript.

Lines 150-155: When comparing in more detail figures 1d and 1e, one can see that whereas the correlation for the 2yr-cycle matches quite well with the 2-3yr filtered observations, the 4yr-cycle experiments show a slightly different evolution than the 3-5yr filtered correlation. This is not mentioned in the manuscript. I wonder if this is due to the simulation setup.

Response: Thank you for your valuable comment. For a 4-yr ENSO-forced experiment, more disturbance from the stochastic forcing would be expected in the relatively longer time evolution of ENSO transition compared to a 2-yr ENSO-forced experiment. As a result, the most obvious difference between 4-yr experiment and 3-5yr filtered correlation appear around the 12-month lead and 12-month lag with respect to ENSO mature winter, when the ENSO condition is near-zero. We have added some discussion about the difference between 4-yr ENSO-forced experiments with 3-5yr filtering correlation in L154-157 in the revised manuscript.